# Attention Should Be Paid to Adolescent Girl Anemia in China: Based on China Nutrition and Health Surveillance (2015–2017)

**DOI:** 10.3390/nu14122449

**Published:** 2022-06-13

**Authors:** Shujuan Li, Liyun Zhao, Dongmei Yu, Hongyan Ren

**Affiliations:** 1NHK Key Laboratory of Trace Element Nutrition, National Institute for Nutrition and Health, Chinese Center for Disease Control and Prevention, Beijing 100050, China; lisj@ninh.chinacdc.cn (S.L.); zhaoly@ninh.chinacdc.cn (L.Z.); 2State Key Laboratory of Resources and Environmental Information System, Institute of Geographic Sciences and Natural Resources Research, Chinese Academy of Sciences, Beijing 100101, China

**Keywords:** anemia, adolescent girls, spatial disparities, influencing factors, China Nutrition and Health Surveillance, China

## Abstract

Adolescent girls are in the key stages of rapid physical and psychological development and have a great demand for iron. Anemia affects adolescent girls’ health, future development, and even the health of their offspring. There has been limited study of adolescent girl anemia at the national and provincial levels in China. We investigated the anemia status of adolescent girls in China based on data from the China Nutrition and Health Surveillance (CNHS, 2015–2017). The anemia prevalence in Chinese adolescent girls aged 10–17 years is 8.5%, with mild anemia and moderate anemia accounting for 65.9 and 31.8%, respectively, and severe anemia accounting for only 2.3%. Significant urban–rural disparities and regional disparities were found in adolescent girl anemia. The anemia prevalence in adolescent girls varied from 3 to 13.4% in different provinces, and 90% of the provinces had anemia prevalence higher than 5%. Having started menstruating (OR = 2.58, *p* < 0.01) and living in rural areas (OR = 1.18, *p* < 0.05) were risk factors for anemia; having a mother with higher education was a protective factor (OR = 0.87, *p* < 0.05). As for food intake, consuming meat ≥35 g per day was a protective factor (OR = 0.90, *p* < 0.05). Consuming vegetables ≥3 times per day was also a protective factor (OR = 0.72, *p* < 0.01), while consuming vegetables ≥400 g per day was a risk factor (OR = 1.24, *p* < 0.01). Special attention should be paid to adolescent girls, especially to those already having started menstruating, living in rural areas, to those whose mother has a low education level, and to those with a relatively unbalanced diet. Comprehensive measures, including paying special attention to vulnerable areas and vulnerable subgroups of adolescent girls, would reduce the risk of anemia.

## 1. Introduction

Anemia is an important public health problem affecting 1.93 billion people in the world [1]. There are about 273 million children, 496 million non-pregnant women, and 32 million pregnant women threatened by anemia in the world [2], especially in low- and middle-income counties [3]. Adolescent girls are in the key stages of rapid physical and psychological development, and rapid physical growth and the onset of menstruation during adolescence may increase anemia risk greatly. Anemia may affect girls’ future development and fertility and may even influence pregnancy outcome and the health state of the next generation [4].The Second International Conference on Nutrition in Rome in 2014 approved an action plan for maternal, infant, and child nutrition, with a commitment to halve anemia prevalence in women of reproductive age by 2025 (from 2011 levels) [5]. To achieve this goal, detailed information is needed about the hemoglobin and anemia states of women to take targeted measures to reduce anemia.

In China, anemia prevalence in adolescent girls aged 12 to 17 years old was 9.1% in 2010–2013 and 10.4% in 2016–2017 according to the Report on Nutrition and Chronic Diseases of Chinese Residents [6,7], which showed a slight upward trend. According to Liu et al., anemia prevalence in adolescent girls aged 12 and 14 years old was 10.6 and 12.3%, respectively, in 2014, and the anemia prevalence in different provinces varied from 3.3 to 25.0%, with Jiangxi, Hunan, Hainan, and Gansu provinces having rates higher than 20% [8]. Due to the imbalanced development of socioeconomic levels in different regions, there were significant regional differences in adolescent anemia prevalence [8,9]. Nevertheless, there has been limited study on anemia in adolescent girls at the national and provincial levels. Regional anemia status would provide useful information for targeted intervention strategies.

Anemia is affected by many factors, including micronutrient deficiencies, chronic infections, and inherited hemoglobin disorders [10]. In China, 70–90% of anemia is related to iron deficiency [11]. Iron deficiency is affected by micronutrient intake and related to influencing factors, such as nutrition status [12,13], dietary intake status [14,15], and socio-economic situation [16]. Detecting the influencing factors may provide us with effective information for creating targeted prevention measures.

In this study, we analyzed the anemia prevalence in adolescent girls at the national and provincial levels and detected the influencing factors for anemia based on the national and provincial representative data of the China Nutrition and Health Surveillance (CNHS, 2015–2017) and made some suggestions for the prevention of anemia in adolescent girls.

## 2. Materials and Methods

### 2.1. Data Source

Data for adolescent girls aged 10–17 years old from China Nutrition and Health Surveillance (CNHS, 2015–2017) was used in this study. CNHS is a national nutrition and health survey of people of all ages, covering 31 provinces in China. It has both national- and provincial-level representativeness. The detailed sample method can be found in the study by Yu et al. [17]. The children’s field survey was taken in 2016–2017, and surveys at 264 points were actually completed. Due to the small number of surveyed adolescent girls in Tibet, the provincial anemia prevalence in Tibet was not calculated in this study.

### 2.2. Survey Method

The content of the survey included inquiry survey, medical examination, laboratory test, and dietary survey (Food Frequency Questionnaire, FFQ, for all adolescents and 3 consecutive days of 24 h dietary recall for one-third of adolescents; FFQ surveyed the food intake frequency and weight for every time in the last month). The surveys were responded to by the parents for children in grades 1–6 in elementary school and by children themselves for children who were older. Fasting venous blood samples were collected in the medical examination, and hemoglobin was detected by Hemocue method (Hemocue 201+ hemoglobin detect meter).

### 2.3. Anemia Definition 

With altitude <1000 m, for girls aged 10 to 11 years old, anemia definition is: hemoglobin value <115 g/L; mild anemia: hemoglobin value ≥110 g/L and <115 g/L; moderate anemia: hemoglobin value ≥80 g/L and <110 g/L; severe anemia: hemoglobin value <80 g/L. For girls aged 12 to 17 years old, anemia definition is: hemoglobin value <120 g/L; mild anemia: hemoglobin value ≥110 g/L and <120 g/L; moderate anemia: hemoglobin value ≥80 g/L and <110 g/L; severe anemia: hemoglobin value <80 g/L [18]. With altitude ≥1000 m, the adjustments were made to the measured hemoglobin concentration according to WHO standard [18].

### 2.4. Influencing Factor Variables

Influencing factors for adolescent girl anemia included personal status (age, nutritional status, dietary status, and menstruation status), family status (parents’ education, parents’ working state, primary caregiver). Girls’ nutrition status was defined by body mass index (BMI = weight/height squared, weight unit as “kg” and height unit as “m”). Classification for underweight, normal, overweight, and obesity was based on the BMI *z*-scores for individuals aged 7–18 years [19]. Menstruation status was defined by having started menstruating or not. Parents’ education level was divided into low education level (including junior high school and below) and high education level (including high school and above). Parents’ working state included out-migrating or not. Primary caregiver was divided into parents, grandparents, and other relatives. Dietary status for every day was calculated from FFQ, and the meat (livestock and poultry meat and its products, animal viscera, and blood) and vegetable (fresh vegetables) intake frequency (times/d) and weight (g/d) were considered.

### 2.5. Statistical Analysis Method

We used SAS 9.4 (SAS Institute Inc., Cary, NC, USA) to analyze the data for adolescent girl anemia and detected its influencing factors. Univariate logistic and multivariate logistic regression were used to analyze the influencing factors of adolescent girl anemia. The data processing procedures are shown in Figure 1.

### 2.6. Mapping Method

ArcGIS 10.7 software was used to map adolescent girl anemia prevalence at provincial and surveillance-point levels.

### 2.7. Ethical Approval

This study was approved by the ethics review committee of National Institute of Nutrition and Health, Chinese Center for Disease Control and Prevention (No. 201614).

## 3. Results

### 3.1. Adolescent Girl Hemoglobin and Anemia Status

In this study, 22,810 adolescent girls aged 10–17 were included (Table 1); 10,700 (46.9%) were in urban areas and 12,110 (53.1%) were in rural areas. The hemoglobin of Chinese adolescent girls aged 10–17 years was 134.68 ± 12.13 g/L. The hemoglobin level was the highest at the age of 12 years (136.08 ± 11.28 g/L) and gradually stabilized at about 133 g/L after the age of 13 years. The anemia prevalence in Chinese adolescent girls aged 10–17 was 8.5%, with mild anemia and moderate anemia accounting for 65.9 and 31.8%, respectively, and severe anemia accounting for only 2.3% (Table 1). The anemia prevalence in rural girls (9.4%) was significantly higher than urban girls (7.6%). The anemia prevalence in girls aged 11 years was the lowest (2.8%), and that of girls aged 17 years was the highest (12.7%). Rural girls aged 15–17 years presented higher anemia prevalence, with anemia prevalence varying from 13.9 to 15.2%.

### 3.2. Regional Disparities of Adolescent Girl Anemia

Adolescent girl anemia prevalence in China presented significant regional disparities. As to surveillance points (Figure 2A), adolescent girl anemia prevalence was 0~34.5%. Briefly, 88 (33.3%) out of 264 surveillance points were at the normal level, while 169 (64.0%) were at the mild level, and 7 (2.7%) were at the moderate epidemic level according to WHO standards (anemia detection rate of ≥40% is defined as having a severe epidemic, 20.0–39.9% as having a moderate epidemic, 5.0–19.9% as having a mild epidemic, and ≤4.9% as normal) [18]. Those surveillance sites with anemia prevalence higher than 20% were mainly distributed in Chongqing, Gansu, Hubei, Sichuan, Shanghai, Shandong, and Hebei provinces. Adolescent girl anemia prevalence at big city surveillance points varied from 2.2 to 13.6%, and small-to-middle-sized cities’ anemia prevalence varied from 0.0 to 34.5%, ordinary rural areas’ anemia prevalence varied from 0.0 to 31.3%, and poverty-stricken rural areas’ anemia prevalence varied from 1.0 to 34.2%.

At the provincial level (Figure 2B), in Ningxia, Beijing, Tianjin and Jilin, adolescent girls’ anemia prevalence was under 5%, while in the other provinces, anemia prevalence was 5.0–13.4%. According to the WHO standard [18], adolescent girls’ anemia rates in most of the provinces in China were at the mild prevalence level. In Anhui, Gansu, Chongqing, and Hainan, adolescent girls had higher anemia prevalence rates of more than 10.0%.

### 3.3. Influencing Factors of Anemia

According to univariate analysis (Appendix A Table A1), having started menstruating, having other relatives as caregivers, parents out-migrating, and vegetable intake ≥400 g/d were risk factors for adolescent girl anemia (*p* < 0.05), while being obese, having parents with higher education levels, and vegetable intake ≥3 times/d were protective factors for anemia (*p* < 0.05).

According to multiple logistic regression results (Table 2), for all adolescent girls, living in rural areas (OR = 1.18, *p* < 0.05), having started menstruating (OR = 2.58, *p* < 0.01), and vegetable intake ≥400 g/d (OR = 1.24, *p* < 0.01) were risk factors for anemia, while having a mother with high education level (OR = 0.87, *p* < 0.05), meat intake ≥35 g/d (OR = 0.90, *p* < 0.05), and vegetable intake ≥3 times/d (OR = 0.72, *p* < 0.01) were protective factors.

As for urban girls, having started menstruating (OR = 2.28, *p* < 0.01) and living in Central China (OR = 1.34, *p* < 0.01) were risk factors, and meat intake ≥35 g/d (OR = 0.85, *p* < 0.05) was a protective factor. For rural girls, having menstruation (OR = 2.85, *p* < 0.01) and vegetable intake ≥400 g/d (OR = 1.28, *p* < 0.05) were risk factors, while being obese (OR = 0.58, *p* < 0.05) and vegetable intake ≥3 times/d (OR = 0.70, *p* = 0.01) were protective factors.

## 4. Discussion

Anemia threatens billions of women and children’s health in the world. It brings great health and economic burdens for countries, especially for developing countries. Adolescent girls are in a key period of nutritional vulnerability due to the increased nutritional demands for growth and development during this phase [20], especially for iron demand due to the onset of menstruation and changes in dietary habits affected by the family and peers [20]. Adolescent girl anemia not only affects individual development but also may have an influence on the next generation. It is important to control and reduce anemia in adolescent girls. In order to provide important information for anemia intervention strategies, national and provincial anemia prevalence and the influencing factors for adolescent girl anemia were analyzed based on data from CNHS (2015–2017).

In 2016–2017, anemia prevalence in Chinese adolescent girls aged 10 to 17 years was 8.5%, with mild and moderate anemia accounting for 97.6%. According to WHO standards [18], it was a mild public health problem. As for girls of different ages, girls aged 15–17 years had a higher anemia prevalence. Globally, non-pregnant women of childbearing aged (15–49 years) had an anemia prevalence of 29%. In high-income regions it was 16%, while in East and Southeast Asia it was 21%, and in Central and West Africa and South Asia the prevalence was highest (47–48%) [2]. Compared with anemia prevalence in adolescent girls aged 10 to 18 years in Korea (5.3%) [21] and that of girls aged 12.0 to 14.99 years old in the United States (3.8%) [22], anemia prevalence in adolescent girls in China was little higher. The results indicated that more attention should be paid to adolescent girl anemia in China.

Spatial disparity results showed that adolescent girl anemia in urban areas was significantly lower than that of rural areas in China (*p* < 0.01). Girls aged 15 to 17 years from rural areas had the highest anemia prevalence (13.9–15.2%). The significant difference between urban and rural adolescent girl anemia indicated the imbalanced development in children [23]. The surveillance-point mapping results showed that there was no big city surveillance point with anemia prevalence higher than 20%, while there were some surveillance points in the small-to-middle-sized cities, ordinary rural areas, and poverty-stricken rural areas having anemia prevalence higher than 20% and even up to 34.5%. These points distributed in Chongqing, Gansu, Hubei, Sichuan, Shanghai, Shandong, and Hebei provinces, which should be paid special attention. At the provincial level, anemia prevalence rates in Beijing, Tianjin, Ningxia, and Jilin were less than 5%, while those of Hainan, Chongqing, Gansu, and Anhui were higher than 10%. Similar findings were reported in the study by Luo et al. [24]. According to the study by Luo et al., anemia rates in girls aged 14 years old in Jiangxi, Hunan, Guangxi, Hainan, and Gansu were higher than 20% in 2014 [24]. In general, the adolescent girl anemia problems in most provinces should be alarming since anemia status is unstable and adolescent girls easily suffer from anemia due to many influencing factors. Special attention should also be paid to the urban girls in the areas of Central China, an area which was a risk factor for anemia in urban girls.

Adolescent girls are in a rapid physical development stage and need more iron intake. However, adolescent girls tend to take in less iron from less or unbalanced food intake and lose more iron from menstruation. As a result, adolescent girls are prone to anemia, which is affected by many individual and family factors. According to the univariate logistic analysis, having started menstruating, having other relatives as caregivers, parents out-migrating, and vegetable intake ≥400 g/d were risk factors for adolescent girl anemia, while being obese, having parents with higher education levels, and vegetable intake ≥3 times/d were protective factors for anemia. Having started menstruating is the most important influencing factor for adolescent girl anemia, which was confirmed by results of the anemia prevalence in girls of ages 12 and 13. According to a study, the age of menarche in Chinese girl is 12.58 years old in urban areas and 12.90 years old in rural areas [25]. Attention should be paid to adolescent girls who have started menstruating and targeted measures, such as supplementary iron and folic acid intake [26], could effectively improve anemia. Caregivers play an important part in adolescent girls’ nutrition and health. Compared with parents and grandparents as caregivers, having other relatives as caregivers was a significant risk factor for anemia, since girls at this age are in a delicate physical and mental development stage and might not receive careful care. At the same time, parents out-migrating was a risk factor due to the reason mentioned above. The results were consistent with study of left-behind children in poverty-stricken rural areas of China [15]. Parents’ education levels also influenced adolescent girl anemia, because the education level might affect the parents’ child-rearing knowledge, as did the family income, which eventually affect children’s nutrition status [27,28].

Adolescent girls’ nutrition status and dietary habits are the direct influencing factors for anemia. Malnutrition includes undernutrition, micronutrient deficiencies, overweight and obesity according to the Second International Conference on Nutrition [5]. Undernutrition, including underweight and stunting, might lead to anemia because the normal growth of adolescents depends on adequate nutrition, and undernutrition is frequently associated with nutrition deficiency and eating disorders [29]. Overweight and obesity were both found to decrease or increase anemia risk. On one hand, overweight and obesity might decrease anemia risk because those overweight or obese people tend to have a better nutritional intake [30]. On the other hand, overweight and obesity might increase the risk of anemia due to imbalanced diets and increased iron requirements due to larger blood volume and/or body size [31,32,33]. In this study, we found that, in rural areas, adolescent girls in the obesity state had lower anemia risk, which was in accordance with other studies in China [34] and in India [30]. The results indicated that in rural areas of China, adolescent girls had a relatively poor nutritional intake, which was confirmed in the following dietary intake status results.

Anemia is considered the most common nutritional deficiency worldwide, and in 95% of cases it is associated with an iron-poor diet [20]; the most important factors responsible for iron deficiency anemia in adolescents are iron-poor diet and inappropriate dietary habits according to a literature review on iron deficiency anemia in adolescents [20]. As for the dietary intake status, our results indicated that meat and vegetable intake frequency and weight affected adolescent girl anemia. According to *Dietary guidelines for school-aged children in China (2022)*, the recommendation of livestock and poultry meat per day for children aged 11–17 years old is 50–75 g, and the recommendation for vegetables per day is 400–500 g [19]. In this study, the weights of meat and vegetable intake were calculated. The weights of meat were as follows: all girls (P25, P50, P75) = 11.9 g, 31.4 g, 67.9 g; urban girls (P25, P50, P75) = 17.0 g, 41.2 g, 84.4 g; rural girls (P25, P50, P75) = 8.6 g, 24.8 g, 53.3 g. The vegetable intake weights were as follows: all girls (P25, P50, P75) = 80 g, 150 g, 260 g; urban girls (P25, P50, P75) =100 g, 200 g, 300 g; rural girls (P25, P50, P75) = 70 g, 130 g, 210 g. According to our results, meat intake ≥35 g/d was a protective factor for adolescent girl anemia. Meat is the main iron source of food, while the adolescent girls’ meat intake was relatively low. Meat intake is still a simple and effective way to improve anemia, which was confirmed in similar studies [6,15,35]. While vegetable intake ≥3 times/d was a protective factor for anemia, vegetable intake ≥400 g/d was a risk factor. Vegetables can provide humans with folic acid, vitamin C, and some iron, but limited iron with low absorptivity is provided by vegetable food. Consuming vegetables more frequently and of varieties was found to be a protective factor for anemia in many studies [15,35]. In this study, we found that vegetable intake of more than 400 g per day was a risk factor for anemia. This could be explained by vegetable intake of more than 400 g per day for adolescent girls being relatively high and the absorptivity of vegetables is low compared with animal products. Similar results were found as an influencing factor in rural adolescent girls’ anemia. According to the Report on Nutrition and Chronic Diseases of Chinese Residents (2020), the average vegetable intake for urban girls aged 12–17 years old is 184.4 g/d, and the intake of livestock and poultry meat is 118.8 g/d. For rural girls’, these values are 164.4 g/d and 91.9 g/d, respectively [7]. Eating relatively too many vegetables may not equate to a balanced diet for adolescent girls who have a great demand for iron. For adolescent girls, a balanced diet is still a simple but effective way to improve anemia. Awareness of anemia and nutritional counseling to improve the quality of the diet should be applied in adolescent girl anemia improvement measures.

The strengths of this study lie in its large representative data set, both on national and provincial levels, from the China Nutrition and Health Surveillance, which is the national survey following the standardized protocols and data collection procedures by the Chinese Center for Disease Control and Prevention. This study provided not only the current anemia status of adolescent girls but also the regional disparities and influencing factors for anemia, which may serve as a basis for the precise prevention and control of anemia.

Some limitations in this study should be mentioned. Firstly, this study did not distinguish whether anemia was iron deficiency anemia or not, which may affect the influencing factor results, but this problem could be solved by the fact that 70–90% of anemia in China is related to iron deficiency [11]. Secondly, in this study we used FFQ data for dietary analysis instead of the 3 day/24 h dietary survey data due to the fact that not all the adolescent girls followed the 3 day/24 h dietary survey method; the FFQ surveyed the general dietary situation in the past month, which might not accurately reflect the dietary state of adolescent girls. We will explore this information to provide more specific information for anemia improvement in future studies.

## 5. Conclusions

Adolescent girls’ anemia is still an important public health problem that should be paid more attention in China, with anemia prevalence varying from 3 to 13.4%, and 90% of the provinces have anemia prevalence higher than 5%. Special attention should be paid to adolescent girls, especially to those already having started menstruating, living in rural areas, to those whose mother has a low education level, and to those with relatively unbalanced diets.

## Figures and Tables

**Figure 1 nutrients-14-02449-f001:**
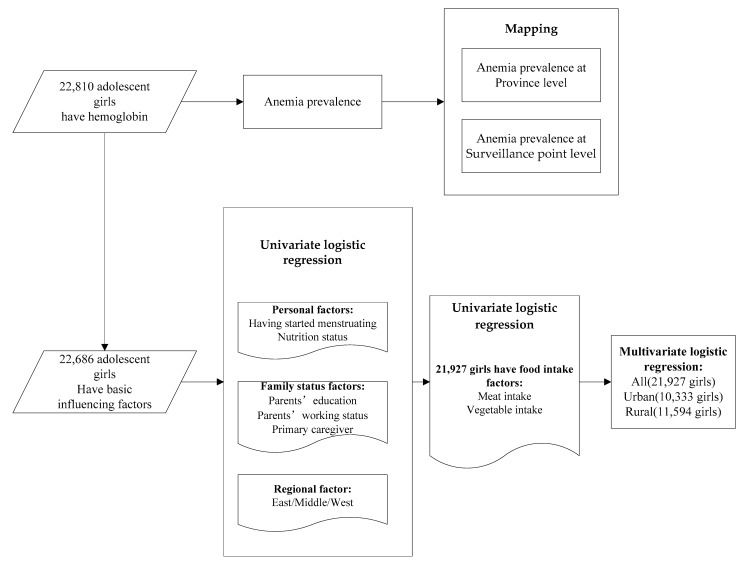
The flowchart of study procedures.

**Figure 2 nutrients-14-02449-f002:**
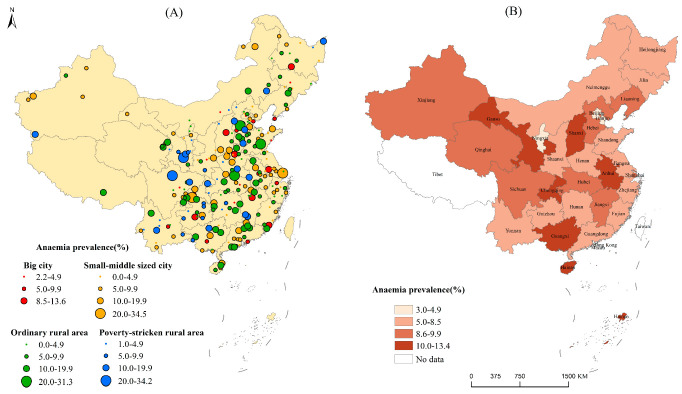
(**A**) Anemia prevalence in adolescent girls at surveillance-point level; (**B**) anemia prevalence in adolescent girls at provincial level.

**Table 1 nutrients-14-02449-t001:** Hemoglobin and anemia status of adolescent girls.

Characteristic	N	Hb(g/L)	Anemia%					
			All (95%CI)	Mild	Moderate	Severe	Urban	Rural
Total	22,810	134.68 ± 12.13	8.5 (7.4–9.7)	5.6	2.7	0.2	7.6 **	9.4 **
Child’s age (in year)								
10	3533	134.46 ± 10.65	4.0 (1.3–6.7)	2.1	1.8	0.1	5.5	3.0
11	3564	136.02 ± 10.46	2.8 (1.3–4.3)	1.6	1.2	0.0	2.9	2.8
12	3646	136.08 ± 11.28	6.0 (4.4–7.7)	4.8	1.1	0.2	4.5	7.4
13	3234	134.86 ± 12.32	10.3 (7.8–12.7)	7.1	3.1	0.1	8.5	11.8
14	1830	134.73 ± 13.02	9.7 (7.3–12.1)	7.0	2.4	0.3	10.9	8.9
15	2426	132.92 ± 12.95	11.6 (9.3–13.8)	7.2	4.2	0.2	9.4 **	13.9 **
16	3205	133.42 ± 13.71	11.5 (9.4–13.7)	7.8	3.4	0.4	9.2	14.5
17	1372	133.57 ± 13.92	12.7 (9.9–15.4)	8.1	4.5	0.1	10.3 **	15.2 **

** represented *p* < 0.01.

**Table 2 nutrients-14-02449-t002:** Factors associated with adolescent girls’ anemia based on multivariate logistic regression analyses.

Model	Influencing Factors	Reference	OR(95%CI)	*p*
All girls	Urban/Rural			
	Rural	Urban	1.18 (1.05–1.31)	0.04
	Having started menstruating			
	Yes	No	2.58 (2.28–2.92)	<0.01
	Mother’s educational level			
	High	Low	0.87 (0.76–0.99)	0.03
	Meat intake weight			
	≥35 g/d	<35 g/d	0.90 (0.81–1.00)	0.04
	Vegetable intake weight			
	≥400 g/d	<400 g/d	1.24 (1.07–1.43)	<0.01
	Vegetable intake frequency			
	≥3 times/d	<3 times/d	0.72 (0.58–0.89)	<0.01
Urban girls	Having started menstruating			
	Yes	No	2.28 (1.89–2.76)	<0.01
	Distribution			
	Middle	East	1.34 (1.11–1.62)	<0.01
	West	East	1.17 (0.97–1.41)	0.11
	Meat intake weight			
	≥35 g/d	<35 g/d	0.85 (0.73–1.00)	0.04
Rural girls	Having started menstruating			
	Yes	No	2.85 (2.42–3.36)	<0.01
	Nutrition status			
	Underweight	Normal	1.15 (0.86–1.52)	0.34
	Overweight	Normal	0.84 (0.65–1.08)	0.16
	Obesity	Normal	0.58 (0.36–0.92)	0.02
	Vegetable intake weight			
	≥400 g/d	<400 g/d	1.28 (1.04–1.57)	0.02
	Vegetable intake frequency			
	≥3 times/d	<3 times/d	0.70 (0.53–0.92)	0.01

A stepwise method of variable entry was used.

## Data Availability

The data is not allowed to be disclosed according to the National Institute for Nutrition and Health and the Chinese Center for Disease Control and Prevention.

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
