# Peer review of "Attention Should Be Paid to Adolescent Girl Anemia in China: Based on China Nutrition and Health Surveillance (2015–2017)"

_nutrients, 2022, doi:10.3390/nu14122449_

Round 1
Reviewer 1 Report
The conclusion that "Balanced diet including enough meat and vegetables would reduce the risk of adolescent girl anaemia" should be revised.
In this sense, the assertion regarding meat consumption and reduced risk of anaemia needs to be proved in several respects:
First. It has only been seen as an associated risk factor in one part of the sample studied, the urban population group.
Second. While data are provided on the number of times vegetables and meat are consumed per week, no information is provided on the size of the portion consumed and its weight in grams. In the case of meat, this could be of great interest in order to be able to relate it to the evidence regarding the recommendation of meat consumption and the risk of suffering from anaemia and, above all, to know if there are differences between the size and weight (in grams) of the portions of food consumed by both groups (urban and rural). If these data are available (especially in the case of meat), they should be included in the results and discussion.
Finally, the inclusion in the conclusions of the recommendation to eat meat to reduce the risk of anaemia should be tested against the evidence with robust studies cited in the discussion.
Reviewer 2 Report
1. The manuscript needs careful editing for style, English usage, and organization of the flow of the methods, results, and discussion.
2. Statistical methods need some detail.
3. Conclusions need to focus on or be relevant to the main results.
4. References need to be reformatted according to Nutrients authors’ instructions. Complete/correct some references (e.g., #5, #6, #7, #18, #24).
Round 2
Reviewer 2 Report
Thanks for addressing my concerns. I have no more questions, except that I think some references are still malformed (e.g., #1, #4, #5, and #15 If possible, please make corrections according to the Nutrients author's instructions.